# RECOVERING OF THE ABSORPTION PARAMETERS IN ACOUSTIC TOMOGRAPHY BY TWO-STAGES DATA PROCESSING

## ABSTRACT

This work is dedicated to the reconstruction of the acoustic parameters of the medium, in particular, the acoustic absorption, by using the pressure, registered , from the gradients calculated to solve the inverse problem. The spatial gradients and the target images of the medium density are interpreted as input and output images, respectively.

## 1 INTRODUCTION

In this paper the deep learning approach to acoustic tomography is considered. The ultrasound tomography was studied intensively lately Duric et al. (2012); Jifik et al. (2012); Wiskin et al. (2017; 2019b;a); Burov et al. (2015). From the mathematical point of view such problems are formulated in form of an coefficient inverse problems Kabanikhin (2008), where acoustic parameters of the medium are to be found by using additional information Shurup (2022). Therefore it is very important to implement the effective techniques for solving the inverse problem due to their connected efficiency.

However, numerical solution of such problems can be difficult due to several reasons, and one of the major difficulties is that solution of inverse problems in 2D and 3D requires significant computational resources. One of the possible ways to overcome such obstacle is to use the data-driven approaches for inverse problem solution Arridge et al. (2019); Aspri et al. (2020); Li et al. (2020); Jin et al. (2020); Bao et al. (2020); Obmann et al. (2020); Stephan & Haltmeier (2021). Rapid development of the deep learning algorithms in the last decade encourages researchers in different fields to study them in order to provide more efficient methods for solving problem of all kinds.

As for the current paper, this work is the direct continuation of the research, provided in Prikhodko et al. (2024). However, in this work we focus on two new aspects. First, we aim to study the possibility of recovering acoustic absorption of the inclusions. It was shown Shishlenin et al. (2021), that the absorption (or acoustic attenuation) has relatively weak influence on the data and is hard to recover when using model-driven approaches. Also we provide new structure of the neural network. On the first step, we use the mathematical model, which describes the process, to compute the gradient of the cost functional. Then we construct the network that can connect this gradient to the desired absorption distribution. Thus, we use physically accurate model to convert a series of 1d pressure curves into a gradient, which has the same dimension as a desired function.

As a model we use the hyperbolic first-order system to describe the propagation of the acoustic waves Kabanikhin (2020). On the one hand, this allows to propose a more realistic model from the physical point of view, when compared to methods, based on the wave equation. On the other hand, we can apply an optimization approach for recovering coefficients of such system, like the density of the medium, the speed of the wave propagation or the absorption coefficient. The mathematical model of acoustic tomography is described by a first-order hyperbolic system PDE and is based on conservation laws. This model guarantees us that the training sets of dynamic data are close to the physical solution.

Now let us highlight modern results of applying deep learning in tomography and related fields. For the sake of brevity we do not aim to mention all the related results and describe only a handful of related works.

It has been demonstrated that group equivariant convolutional operations can naturally be included in the learned reconstruction methods for inverse problems, which are motivated by the variational regularization approach Celledoni et al. (2021). The learned iterative methods were developed in which proximal operators were modeled as group equivariant CNNs. The developed method was applied to the reconstruction of low-dose computed tomography and reconstruction of magnetic resonance imaging with subsampling.

Full Wave Inversion (FWI) approach with coupled control and deep learning (AD-DLFWI) was developed Zhang et al. (2021), which uses a fully convolutional network (FCN) to invert subsurface velocity from reflected seismic data. In the proposed approach, a seismic image of the application of the adjoint operator of the scattering wave equation, which is equivalent to the gradient of the classical FWI, was used as the FCN data processing mechanism. FWI in acoustic tomography also are considered in Matthews et al. (2017); Fu Li (2012), FWI and DL in Feigin et al. (2019; 2020a;b).

It was investigated DL in tomography and medical imaging Araya-Polo et al. (2018; 2020); Denker et al. (2021); Lozenski et al. (2023), CNN in ultrasound tomography Zhao et al. (2020), DNN in breast ultrasound Jush et al. (2020).

DNNs framework for predicting spatially varying crystallographic orientations in a weld based on ultrasonic time data was presented Singh et al. (2022).

A deep neural network which accounts for a nonlinear forward operator and primal-dual algorithm by network architecture Fan et al. (2022) was designed for fast image reconstruction algorithm for ultrasound computed tomography. The network was trained with ultrasound pressure field data as input to get directly an optimized reconstruction of speed of sound and attenuation images. The training and test data were based on a set of Optical and Acoustic Breast Phantom Database, where it was used the image as truth to simulate pressure field data.

The experimental results show that proposed deep learning methods help to improve the neural networks' robustness against noise and the generalizability to real measurement data Zhao et al. (2023).

An overview of the technologies used to obtain breast images and algorithms used to detect breast cancer was presented Basurto-Hurtado et al. (2022).

The mathematical design of a new gesture-based input/command device with identification and prediction using a moving emitter was proposed based on parameter inversion model by neural network to reconstruct the trajectory of the moving point source Zhang et al. (2023).

## 2  MATHEMATICAL MODEL AND FIRST PROCESSING OF THE DATA

First let us describe the general setting of the problem. Based on a current trends and prototypes, we considered the general formulation of the problem according to the figure 1. The acoustic wave source (1) generates a probing pulse (2), which passes through the inner area of the tomograph filled with water (3) and is scattered on the studied object (4), possibly containing inclusions of the type (5). The problem is to locate inclusions and their characteristics based on acoustic pressure, measured in receivers (8). The measured acoustic pressure in the receivers contains information about both the transmitted wave through the body under study and the reflected wave. Both source and receivers are located on the boundary of the tomograph.

We use the mathematical model, based on the first order PDE system, to simulate the mentioned scheme. We consider the following first order system of hyperbolic equations based on the conservation laws Klyuchinskiy et al. (2021a):

$$\frac{\partial u}{\partial t} + \frac{1}{\rho}\frac{\partial p}{\partial x} = 0, \qquad \frac{\partial v}{\partial t} + \frac{1}{\rho}\frac{\partial p}{\partial y} = 0, \quad (x,y) \in \Omega, \quad 0 < t \le T, \tag{1}$$

$$\frac{\partial p}{\partial t} + \sigma p + \rho c^2 \left( \frac{\partial u}{\partial x} + \frac{\partial v}{\partial y} \right) = \theta_s(x,y)I(t), \qquad (x,y) \in \Omega, \tag{2}$$

$$u, v, p|_{t=0} = 0 \tag{3}$$

with non-reflected boundary conditions. Here $u = u(x,y,t)$, $v = v(x,y,t)$ are components of velocity vector, with respect to $x$ and $y$ respectively, $p = p(x,y,t)$ is the acoustic pressure. The

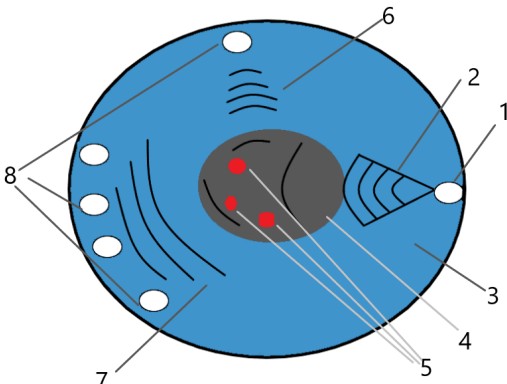

Figure 1: The model of acoustic tomograph: 1 - Border of the tomograph, covered with absorptive material, 2 - liquid inside the tomograph, 3 - source of the acoustic waves, 4 - object of investigation, 5 - inclusions, 6 - reflected wave, 7 - passed wave, 8 - receivers.

parameters of the system describes the properties of the medium: $\rho(x, y)$ denotes the density of the medium, and $c(x, y)$ is the speed of the wave propagation, $\sigma(x, y)$ is the absorption (another term is acoustic attenuation). $\Omega = (x, y) \in [0, L] \times [0, L]$, function $\theta_s$ describes the location of the source, $I(t)$ has the form of Ricker wavelet. The data is registered inside receivers:

$$p(x_j, y_j, t) = f_j(t), j = 1, \ldots, N.$$

We assume that each source is in fact so called transducer, that can generate or register data. Thus, the number of total possible sources and number of receivers is the same. Since in this paper we consider the recovering the absorption, then our goal is to compute one 2D function $\sigma(x, y)$ by a number of $N \times N$ one-dimensional pressure functions, registered in the receivers for each possible source location. Now let us consider the auxiliary functions $\boldsymbol{\Psi} = (\Psi_1, \Psi_2)$ and $\Psi_3$, that solves the following problem 1-3:

$$\rho \frac{\partial \boldsymbol{\Psi}}{\partial t} + \nabla \Psi_3 = 0, \tag{4}$$

$$\frac{\partial \Psi_3}{\partial t} + \rho c^2 \nabla \cdot \boldsymbol{\Psi} = 2\rho c^2 \sum_{i=1}^{N} \delta(x - x_i, y - y_i) \big[ p - f_i \big], \tag{5}$$

$$\Psi_j|_{t=T} = 0, j = 1, 2, 3 \tag{6}$$

with non-reflected boundary conditions. The system 4—6 is adjoint to the 1—3 with respect to the following cost functional:

$$J(\sigma) = \int_0^T \sum_{i=1}^{N} \big[ p(x_i, y_i, t; \sigma) - f_i(t) \big]^2 dt \to \min_{\sigma}$$

When the solution of 4-6 is known, one can compute the gradient of the cost functional as follows Shishlenin et al. (2023):

$$J'(\sigma)(x, y) = \int_0^T \frac{p(x, y, t)\Psi_3(x, y, t)}{\rho(x, y)c^2(x, y)} dt. \tag{7}$$

Usually such approach is used in optimization methods of solving inverse problems. However, such scheme usually requires quite a lot of iterations. Even if we use optimization and calculate the gradient and the associated problem at the same time Klyuchinskiy D. (2020); Klyuchinskiy et al. (2021b), as well as apply *a priori* information about the inverse problem solution Kabanikhin & Shishlenin (2008), the time spent on calculations will be long. Thus, we use the formulae 7 only once and use the registered data $f_i(t)$ to find $J'(\sigma)(x, y)$. On the next step we construct the neural network that allows us to recover $\sigma(x, y)$ by computed $J'(\sigma)(x, y)$.

Recently, works have appeared that investigate nonlinear wave propagation processes in acoustic tomography Muir & Carstensen (1980); Wiskin et al. (2017); Meliani & Nikolić (2022); Tabak et al. (2022); Shishlenin et al. (2024).

## 3 Data description

A synthetic dataset was created to train and validate the model. The creation of the synthetic dataset is one of the key steps in this study. The dataset consists of models, each containing one or more inclusions of different shapes and different values of absorption coefficient inside the inclusion. In particular, the inclusion shapes can be triangular or oval. The acoustic parameters of the inclusions were taken close to the parameters of the human body Mast (2000); Okawai et al. (2001).

There are several reasons for choosing synthetic data. First, they allow for uniform coverage of the data domain, which in turn allows for a more accurate assessment of the performance and quality of the model. Secondly, synthetic data provides the ability to generate an almost unlimited number of unique training examples, which contributes to the training of a more versatile and reliable model.

The data generation process includes a series of calculations with solutions to direct and adjoint problems. Each example of the model is randomly generated, their inclusions (shape, size, position and value of absorption inside) are varied to provide a variety of training examples. For each constructed model distribution $\sigma(x, y)$ we solve the direct problem 1-3 to compute the synthetic pressure curves. Then we solve 4-6 to obtain $J'(\sigma)(x, y)$ for each given model.

We should mention, that the proposed approach requires additional computations for each model we solve one direct problem 1—3 and one adjoint problem 4—6. However, the system 4—6 is also the hyperbolic system in reversed time and can be solved by the same means as the direct problem 1—3. Therefore, the CPU cost of calculating the gradient of the functional is somewhat equivalent of two solutions of direct problem. During numerical experiments, we used MUSCL–Hancock method for the direct and adjoint problem solution, which is full time and space second-order accurate Novikov et al. (2021).

After creation, the synthetic dataset is divided into training, validation, and test samples. This separation provides an opportunity to conduct an honest assessment of the model's performance, as well as control over its ability to generalize to new data, such as a new geometry of inclusions.

Within the framework of this study, we interpreted the gradients of the cost functional (the input data) and the acoustic absorption (the output data) as images. This data representation is very convenient, as it allows us to use techniques and algorithms designed specifically for image processing, in particular, convolutional neural networks.

## 4 Architecture

In this study, the architecture of the UNet convolutional neural network is used to solve the problem of restoring the density of the medium by spatial gradients. This architecture was chosen because of its performance in input and output image-related tasks, in particular in image segmentation and reconstruction tasks. The UNet architecture consists of two main parts (an encoder and a decoder), connected by a bottleneck.

### 4.1 Encoder

The encoder consists of a sequence of convolution blocks, each of which contains two convolutional layers with a ReLU activation function and a batch normalization layer after each convolutional layer. The convolution is applied to the input data layer using a filter (convolution kernel) as follows: $(I * K)(i, j) = \sum_m \sum_n I(m, n) K(i - m, j - n)$, where $I$ - the input data layer, $K$ - convolution kernel, $(i, j)$ - coordinates in the output data layer, $(m, n)$ - coordinates in the input data layer. ReLU (Rectified Linear Unit) activation function is applied to each element of the input data layer as follows: $ReLU(x) = \max(0, x)$, where $x$ is the input data layer. Batch normalization is applied to each individual input data layer as follows: $BN(x) = \gamma \left( \frac{x - \mu}{\sqrt{\sigma^2 + \epsilon}} \right) + \beta$, where $x$ is the input data layer, $\mu$ is the mean value of the input layer, $\sigma$ is the variance, $\epsilon$ is auxiliary parameters used for

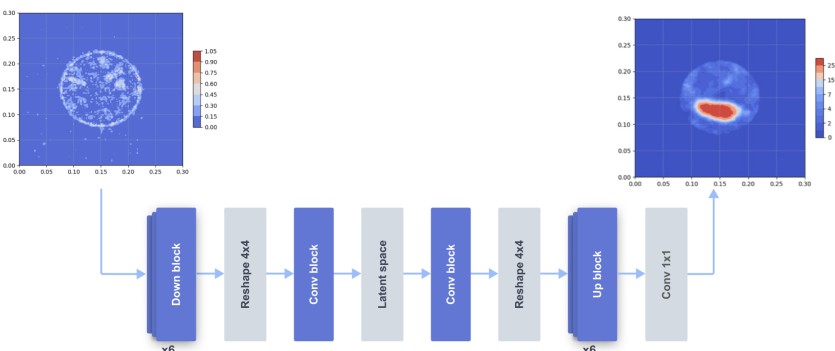

Figure 2: Proposed architecture of of the UNet convolutional network.

the stability of computations, $\beta$ is the shift parameter. Each ConvBlock is applied to the input data, reducing its spatial dimensions and increasing the channel depth. This process continues until the desired depth of the model is reached.

## 4.2 BOTTLENECK

The bottleneck is an intermediate block that processes the data after the encoder and transmits it to the decoder. This block also consists of two convolutional layers with a ReLU activation function and a batch normalization layer after each convolutional layer.

## 4.3 DECODER

The decoder processes the data from the "bottleneck" and increases their spatial dimensions to the size of the original image. For this, a sequence of reverse convolution blocks (ConvTranspose2d) and convolution blocks (ConvBlock) is used. The ConvTranspose2d function performs an operation known as transposed convolution (or deconvolution). It is the reverse of the convolution operation and is used to increase the spatial resolution of an image:

$$((I *_{tr} K)(i, j) = \sum_m \sum_n I(m, n) K(i + m, j + n))$$

During computations transposed convolution also involves inserting zero's complement into the input data and subsequent convolution using a conventional convolution kernel. This provides an increase in the spatial resolution of the output data. Each ConvBlock uses two convolutional layers with a ReLU activation function and a batch normalization layer after each convolutional layer.

We should note that we also used a Max Pool 2d layer to reduce the spatial dimensions of the data in the encoder, and a dropout layer to regularize and prevent overfitting of the model.

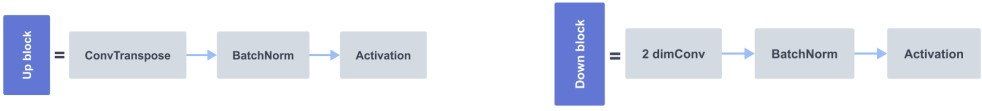

Figure 3: Structure of Up block and Down block

## 5 TRAINING

Model training involves several key steps, which include model initialization, training, and model validation. The model is trained on the training data using the Adam optimizer and the MSE loss

function (RMS error). This process includes a forward pass, where the model makes predictions based on the input data, and a backward pass, where the gradients propagate back through the model. At each stage of training, the loss function is calculated based on the model's predictions and true values. The gradients are then propagated back through the model and the weights of the model are updated using the optimizer. After each training epoch, the model is validated on the validation data. Validation involves going through the model directly and calculating the loss function, but without going through it back or updating the weights. After each epoch, it is checked whether the current model is the best based on validation losses. If the model is the best, then it is saved. At any time after training, you can upload the best model for further use or evaluation.

We also used an early stop to prevent overfitting. If the model does not improve over a certain number of epochs, then training stops. The training schedule can be observed on The learning process took about 4 hours. During this process, 40,000 training examples were used, which allowed the model to obtain enough data for training. The optimal value was found at 524 epochs. An algorithm was also used to reduce the descent parameter when the loss function reaches a plateau. The results of training are presented on the figure 4.

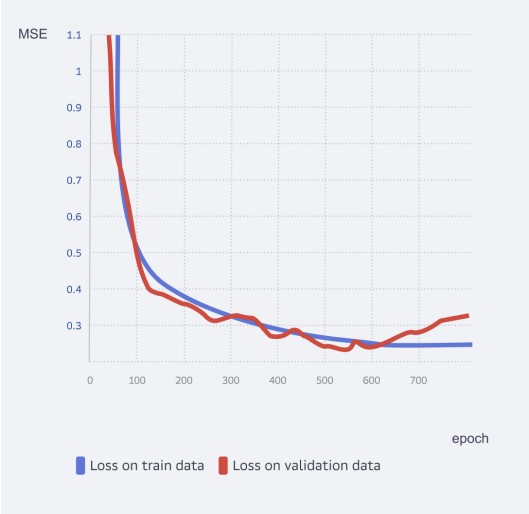

Figure 4: Training results

## 6 COMPUTATION RESULTS

After training the model on a synthetic dataset, testing was performed to assess the quality of the recovering of acoustic absorption. The results are presented in the table 1 and figures 5, 6, 7. The

Table 1: Training results

| Metric | Value |
| --- | --- |
| RMSE | 0.539 |
| MAE | 0.623 |

results show that the model successfully reconstructs the density of the medium from the gradients of space. This can be seen both in the assignment of the metrics that were selected for this task, and in the visualization of the reconstructed images, which demonstrate a high degree of overlap with the original images of the absorption of the medium.

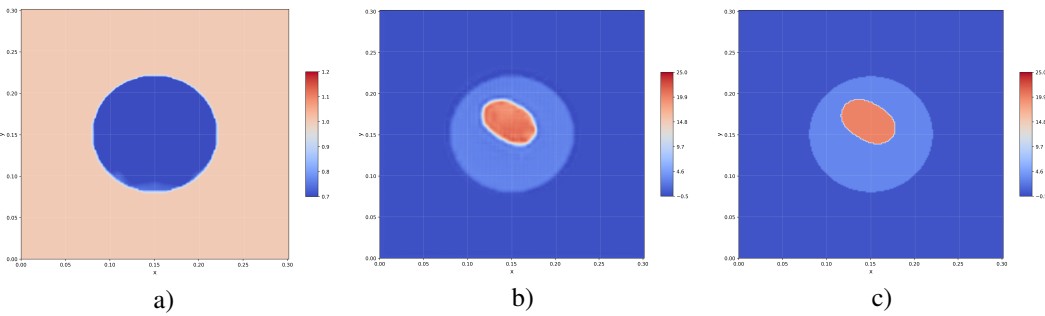

Figure 5: Computational results (convex case): a) - input, b) - computed absorption, c) - true solution.

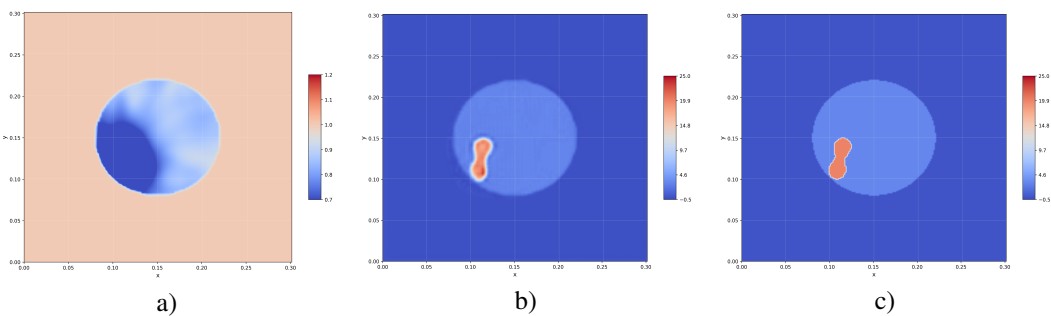

Figure 6: Computational results (structure not presented in dataset): a) - input, b) - computed absorption, c) - true solution.

## 7 CONCLUSION

Visualization of learning allows us to understand more deeply how the model learns and adapts to the data. In this case, we see that the model successfully copes with changing the shape of inclusions that were not present in the training dataset. Figure 6 indicate that the model has a good generalizing ability, which allows it to successfully interpret and adapt to new forms that are not represented in the training dataset. Adding several inclusions of a more complex shape also shows realistic results, which confirms the model's ability to adapt to more complex scenarios. This demonstrates the resilience of the model to various variations in the data. However, the network shows poor performance with non-convex shapes in Fig. 7. This may be due to the fact that non-convex shapes are a more difficult task for the model, requiring more complex patterns to recognize them. This may indicate the need to further improve the model, perhaps by including additional layers or using more complex architectures. The solution obtained by NN approach can be used for classical methods (for example, optimization) as a good initial approximation or *a priori* information.

ACKNOWLEDGMENTS

The work was supported by Russian Science Foundation, project 24-41-04004 "Identification and research of mathematical models in science and industry — regularization and machine learning".

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

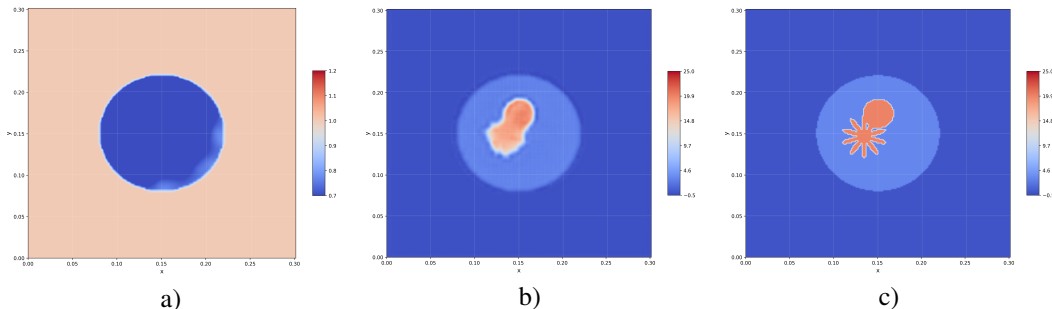

Figure 7: Computational results (non-convex case): a) - input, b) - computed absorption, c) - true solution.

S. Arridge, P. Maass, O. Öktem, and C.-B. Schönlieb. Solving inverse problems using data-driven models. *Acta Numerica*, 28:1–174, 2019. doi: 10.1017/S0962492919000059.

A. Aspri, S. Banert, O. Öktem, and O. Scherzer. A data-driven iteratively regularized landweber iteration. *Numerical Functional Analysis and Optimization*, 41(10):1190–1227, 2020. doi: 10.1080/01630563.2020.1740734. URL https://doi.org/10.1080/01630563.2020.1740734.

G. Bao, X. Ye, Y. Zang, and H. Zhou. Numerical solution of inverse problems by weak adversarial networks. *Inverse Problems*, 36(11), 2020. 115003.

Jesús A. Basurto-Hurtado, Irving Armando Cruz-Albarrán, Manuel Toledano-Ayala, Mario Alberto Ibarra-Manzano, Luis Alberto Morales-Hernández, and Carlos A. Perez-Ramirez. Diagnostic strategies for breast cancer detection: From image generation to classification strategies using artificial intelligence algorithms. *Cancers*, 14, 2022. URL https://api.semanticscholar.org/CorpusID:250657973.

V. A. Burov, D. I. Zotov, and O. D. Rumyantseva. Reconstruction of the sound velocity and absorption spatial distributions in soft biological tissue phantoms from experimental ultrasound tomography data. *Acoustical Physics*, 61:231–248, 2015.

E. Celledoni, M.J. Ehrhardt, C. Etmann, B. Owren, C.-B. Schönlieb, and F. Sherry. Equivariant neural networks for inverse problems. *Inverse Problems*, 37(1), 2021. 085006.

A. Denker, M. Schmidt, J. Leuschner, and P. Maass. Conditional invertible neural networks for medical imaging. *Journal of Imaging*, 7(11), 2021. 243.

N. Duric, P. Littrup, C. Li, O. Roy, S. Schmidt, R. Janer, X. Cheng, J. Goll, O. Rama, L. Bey-Knight, and W. Greenway. Breast ultrasound tomography: bridging the gap to clinical practice. In Johan G. Bosch and Marvin M. Doyley (eds.), *Medical Imaging 2012: Ultrasonic Imaging, Tomography, and Therapy*, volume 8320, pp. 83200O. International Society for Optics and Photonics, SPIE, 2012. doi: 10.1117/12.910988. URL https://doi.org/10.1117/12.910988.

Yuling Fan, Hongjian Wang, Hartmut Gemmeke, Torsten Hopp, and Juergen Hesser. Model-data-driven image reconstruction with neural networks for ultrasound computed tomography breast imaging. *Neurocomputing*, 467:10–21, 2022. ISSN 0925-2312, 1872-8286. doi: 10.1016/j.neucom.2021.09.035. 54.12.03; LK 01.

M. Feigin, D. Freedman, and B.W. Anthony. A deep learning framework for single-sided sound speed inversion in medical ultrasound. *IEEE Trans. Biomed. Eng.*, 67:1142–1151, 2019.

Micha Feigin, Yizhaq Makovsky, Daniel Freedman, and Brian W. Anthony. *High-frequency full-waveform inversion with deep learning for seismic and medical ultrasound imaging*, pp. 3492–3496. 2020a. doi: 10.1190/segam2020-3426935.1. URL https://library.seg.org/doi/abs/10.1190/segam2020-3426935.1.

Micha Feigin, Manuel Zwecker, Daniel Freedman, and Brian W. Anthony. Detecting muscle activation using ultrasound speed of sound inversion with deep learning. In *2020 42nd Annual International Conference of the IEEE Engineering in Medicine and Biology Society (EMBC)*, pp. 2092–2095, 2020b. doi: 10.1109/EMBC44109.2020.9175237.

Nebojsa Duric Mark A. Anastasio Fu Li, Umberto Villa. 3d full-waveform inversion in ultrasound computed tomography employing a ring-array. *Proc. SPIE 12470, Medical Imaging 2023: Ultrasonic Imaging and Tomography*, 12470:705–713, 2012.

Radovan Jifik, Igor Peterlík, Nicole V. Ruiter, Jan Fousek, Robin Dapp, Michael Zapf, and Jiff Jan. Sound-speed image reconstruction in sparse-aperture 3-d ultrasound transmission tomography. *IEEE Transactions on Ultrasonics, Ferroelectrics, and Frequency Control*, 59:254–264, 2012. URL https://api.semanticscholar.org/CorpusID:23026579.

Y. Jin, D. Jiang, and M. Cai. 3d reconstruction using deep learning: a survey. *Communications in Information and Systems*, 20(4):389–413, 2020.

Farnaz Khun Jush, Markus Biele, Peter Michael Dueppenbecker, Oliver Schmidt, and Andreas Maier. Dnn-based speed-of-sound reconstruction for automated breast ultrasound. In *2020 IEEE International Ultrasonics Symposium (IUS)*, pp. 1–7, 2020. doi: 10.1109/IUS46767.2020.9251579.

Klyuchinskiy D.V. Novikov N.S. Shishlenin M.A. Kabanikhin, S.I. Numerics of acoustical 2d tomography based on the conservation laws. *J. Inverse Ill-Posed Probl.*, 28:287–297, 2020.

S. I. Kabanikhin. Definitions and examples of inverse and ill-posed problems. *J. Inverse Ill-Posed Problems*, 16(4):317–357, 2008.

S.I. Kabanikhin and M.A. Shishlenin. Quasi-solution in inverse coefficient problems. *Journal of Inverse Ill-Posed Problems*, 16(7):705–713, 2008.

D. Klyuchinskiy, N. Novikov, and M. Shishlenin. Recovering density and speed of sound coefficients in the 2d hyperbolic system of acoustic equations of the first order by a finite number of observations. *Mathematics*, 9(2), 2021a. 199.

D.V. Klyuchinskiy, N.S. Novikov, and M.A. Shishlenin. Cpu-time and ram memory optimization for solving dynamic inverse problems using gradient-based approach. *Journal of Computational Physics*, 439, 2021b. 110374.

Shishlenin M.A. Klyuchinskiy D., Novikov N. Modification of gradient descent method for solving coefficient inverse problem for acoustics equations. *Computation*, 8(3), 2020.

H. Li, J. Schwab, S. Antholzer, and M. Haltmeier. Nett: solving inverse problems with deep neural networks. *Inverse Problems*, 36, 2020. 065005.

Luke Lozenski, Hanchen Wang, Brendt Wohlberg, Umberto Villa, and Youzuo Lin. Data driven methods for ultrasound computed tomography. In Lifeng Yu, Rebecca Fahrig, and John M. Sabol (eds.), *Medical Imaging 2023: Physics of Medical Imaging*, volume 12463, pp. 124630Q. International Society for Optics and Photonics, SPIE, 2023. doi: 10.1117/12.2654442. URL https://doi.org/10.1117/12.2654442.

T. D. Mast. Empirical relationships between acoustic parameters in human soft tissues. *Acoustics Research Letters*, 1(2):37–42, 2000.

T.P. Matthews, K. Wang, C. Li, N. Duric, and M.A. Anastasio. Regularized dual averaging image reconstruction for full-wave ultrasound computed tomography. *IEEE Transactions on Ultrasonics, Ferroelectrics, and Frequency Control*, 64(5):811–825, 2017.

M. Meliani and V. Nikolić. Analysis of general shape optimization problems in nonlinear acoustics. *Applied Mathematics and Optimization*, 86:39, 2022.

T.G. Muir and E.L. Carstensen. Prediction of nonlinear acoustic effects at biomedical frequencies and intensities. *Ultrasound in Medicine and Biology*, 6(4):345–357, 1980.

N.S. Novikov, D.V. Klyuchinskiy, M.A. Shishlenin, and S.I. Kabanikhin. On the modeling of ultrasound wave propagation in the frame of inverse problem solution. *Journal of Physics: Conference Series*, 2099, 2021. doi: 10.1088/1742-6596/2099/1/012044. 012044.

D. Obmann, J. Schwab, and M. Haltmeier. Deep synthesis network for regularizing inverse problems. *Inverse Problems*, 37(1), 2020. doi: 10.1088/1361-6420/abc7cd. 015005.

H. Okawai, K. Kobayashi, and S. Nitta. An approach to acoustic properties of biological tissues using acoustic micrographs of attenuation constant and sound speed. *Journal of Ultrasound in Medicine*, 20:891–907, 2001.

A. Prikhodko, M. Shishlenin, N. Novikov, and D. Klyuchinskiy. Encoder neural network in 2d acoustic tomography. *Applied and Computational Mathematics*, 23(1):83–98, 2024. doi: 10.30546/1683-6154.23.1.2024.83.

M. Shishlenin, A. Kozelkov, and N. Novikov. Nonlinear medical ultrasound tomography: 3d modeling of sound wave propagation in human tissues. *Mathematics*, 12(2), 2024. ISSN 2227-7390. doi: 10.3390/math12020212. URL `https://www.mdpi.com/2227-7390/12/2/212`.

M.A. Shishlenin, N.S. Novikov, and D.V. Klyuchinskiy. On the recovering of acoustic attenuation in 2d acoustic tomography. *Journal of Physics: Conference Series*, 2099, 2021. doi: 10.1088/1742-6596/2099/1/012046. 012046.

M.A. Shishlenin, N.A. Savchenko, N.S. Novikov, and D.V. Klyuchinskiy. On the reconstruction of the absorption coefficient for the 2d acoustic system. *Siberian Electronic Mathematical Reports*, 20(2):1474–1489, 2023.

A.S. Shurup. Numerical comparison of iterative and functional-analytical algorithms for inverse acoustic scattering. *Eurasian Journal of Mathematical and Computer Applications*, 10(1):79–99, 2022.

Jonathan Singh, Katherine Tant, Anthony Mulholland, and Charles MacLeod. Deep learning based inversion of locally anisotropic weld properties from ultrasonic array data. *Applied Sciences*, 12(2), 2022. ISSN 2076-3417. doi: 10.3390/app12020532. URL `https://www.mdpi.com/2076-3417/12/2/532`.

A. Stephan and M. Haltmeier. Discretization of learned nett regularization for solving inverse problems. *Journal of Imaging*, 7(11), 2021. 239.

G. Tabak, M.L. Oelze, and A.C. Singer. Effects of acoustic nonlinearity on communication performance in soft tissues. *The Journal of the Acoustical Society of America*, 152(6):3583, 2022.

J. W. Wiskin, D. T. Borup, E. Iuanow, J. Klock, and Mark W. Lenox. 3-d nonlinear acoustic inverse scattering: Algorithm and quantitative results. *IEEE Transactions on Ultrasonics, Ferroelectrics, and Frequency Control*, 64(8):1161–1174, 2017. doi: 10.1109/TUFFC.2017.2706189.

James Wiskin, Bilal Malik, Rajni Natesan, David Borup, Nasser Pirshafiey, Mark Lenox, and John Klock. Full wave 3d inverse scattering transmission ultrasound tomography: Breast and whole body imaging. In *2019 IEEE International Ultrasonics Symposium (IUS)*, pp. 951–958, 2019a. doi: 10.1109/ULTSYM.2019.8925778.

James Wiskin, Bilal Malik, Rajni Natesan, and Mark Lenox. Quantitative assessment of breast density using transmission ultrasound tomography. *Medical Physics*, 46(6):2610–2620, 2019b. doi: https://doi.org/10.1002/mp.13503. URL `https://aapm.onlinelibrary.wiley.com/doi/abs/10.1002/mp.13503`.

Ping Zhang, Pinchao Meng, Weishi Yin, and Hongyu Liu. A neural network method for time-dependent inverse source problem with limited-aperture data. *Journal of Computational and Applied Mathematics*, 421:114842, 2023. ISSN 0377-0427. doi: https://doi.org/10.1016/j.cam.2022.114842.

W. Zhang, J. Gao, Z. Gao, and H. Chen. Adjoint-driven deep-learning seismic full-waveform inversion. *IEEE Transactions on Geoscience and Remote Sensing*, 59(10):8913–8932, 2021.

W. Zhao, H. Wang, H. Gemmeke, K.W.A. van Dongen, T. Hopp, and J. Hesser. Ultrasound transmission tomography image reconstruction with a fully convolutional neural network. *Physics in Medicine and Biology*, 65(23):235021, nov 2020. doi: 10.1088/1361-6560/abb5c3. URL `https://dx.doi.org/10.1088/1361-6560/abb5c3`.

Wenzhao Zhao, Yuling Fan, Hongjian Wang, Hartmut Gemmeke, Koen W A van Dongen, Torsten Hopp, and Jürgen Hesser. Simulation-to-real generalization for deep-learning-based refraction-corrected ultrasound tomography image reconstruction. *Physics in Medicine and Biology*, 68 (3):035016, jan 2023. doi: 10.1088/1361-6560/acaeed. URL `https://dx.doi.org/10.1088/1361-6560/acaeed`.

