# OpenReview forum: "Recovering of the absorption parameter in acoustic tomography by two-stages data processing"
_mathai.club/MathAI/2025/Conference — MathAI 2025 Oral_

### Official Review · Reviewer_4f72 · 2025-02-23
**Review on the paper: Recovering of the absorption parameter in acoustic tomography by two-stages data processingRecovering of the absorption parameter in acoustic tomography by two-stages data processing**

**Rating:** 7
**Confidence:** 5

**Review:**

Strong points:
1. Paper provides original approach for inversion of acoustic absorption in seismic tomography.
2. Paper clearly describes the approach and provide experimental results obtained on synthetic data.

Weak points:
1. Paper does not explain practical applications of the method (can be improved)
2. Paper does not address the impact of energy scattering impact on the wave field which may affect practical applicability of the method based on specific applications. See discussion on absorption vs. scattering based on wave length and size of the target object in the papers and dissertation referenced on the pages: http://webstructor.net/geotomo/GEOTOMO/articles.htm and http://webstructor.net/geotomo/GEOTOMO/ru/articles.htm
3. Paper does not provide experimental results on real field data
4. Paper does not compare results with alternative approaches
5. Paper does not have ACKNOWLEDGMENTS section filled properly
 (MUST be fixed for acceptance!)
5. Paper does not have APPENDIX section filled properly
 (MUST be fixed for acceptance!)

---

### Official Review · Reviewer_7vtT · 2025-02-25
**RECOVERING OF THE ABSORPTION PARAMETER IN ACOUSTIC TOMOGRAPHY BY TWO-STAGES DATA PROCESSING**

**Rating:** 7
**Confidence:** 4

**Review:**

Overall Assessment
The article presents an interesting and relevant study focused on recovering acoustic parameters of a medium, particularly the absorption coefficient, using deep learning methods. The work combines rigorous mathematical approaches with modern machine learning techniques, making it significant for the scientific community. The article is well-structured, and the presented results demonstrate the potential of the proposed approach. However, there are several aspects that require refinement or clarification.
Strengths of the Article:
1.	Relevance of the Topic:
-Recovering acoustic parameters, such as absorption, is an important task in medical diagnostics, geophysics, and other fields. The proposed approach could significantly improve the accuracy and speed of solving such problems.
2.	Use of Deep Learning:
-	The authors propose an innovative method that combines physically accurate mathematical models with neural networks. This allows for the consideration of complex physical processes while leveraging the advantages of data-driven approaches.
3.	Synthetic Data:
-	The creation of a synthetic dataset for training and testing the model is a strong point of the work. It enables control over data quality and assessment of the model's generalization ability.
4.	UNet Architecture:
-	The use of the UNet architecture, which has proven effective in image processing tasks, is a justified choice. The authors provide a detailed description of the network's structure, making the work reproducible.
5.	Results:
-	The presented results show that the model successfully recovers absorption parameters for various inclusion shapes. This confirms the practical applicability of the proposed method.

Weaknesses and Remarks:
1.	Limitations of Synthetic Data:
-	While synthetic data is useful for training, its use may limit the model's applicability to real-world data. The authors should consider validating the model on experimental or real-world data.
2.	Issues with Non-Convex Shapes:
-	The authors note that the model struggles with non-convex shapes. This is an important limitation that requires further investigation. It might be worth exploring more complex neural network architectures or additional data preprocessing methods.
3.	Lack of Comparison with Other Methods:
-	The article does not provide a comparison of the proposed method with other approaches, such as traditional optimization methods or alternative neural network architectures. This could strengthen the argument for the method's effectiveness.
4.	Computational Complexity:
-	The authors mention that solving inverse problems in 2D and 3D requires significant computational resources. However, the article does not discuss how the proposed method addresses this issue. An analysis of computational complexity and training time would be beneficial.
5.	Insufficient Visualization:
-	Although the authors provide some graphs, the visualization of results could be more detailed. For example, additional examples of parameter recovery for various inclusion types and comparisons with ground truth values would be helpful.
6.	The article is not formatted according to the template in terms of the number of pages. Only 8 pages.
Recommendations for Improvement:
1.	Incorporate Experimental Data:
-	To increase confidence in the method, it is recommended to validate it on real-world data, such as data obtained from medical or geophysical studies.
2.	Improve Handling of Non-Convex Shapes:
-	It would be worthwhile to explore more complex neural network architectures or additional preprocessing methods to improve results for non-convex shapes.
3.	Conduct Comparisons with Other Methods:
-	Adding comparisons with traditional inverse problem-solving methods or alternative neural network architectures would enhance the scientific value of the work.
4.	Include Computational Complexity Analysis:
-	An analysis of the model's training time and performance on different hardware platforms would be useful.
5.	Enhance Visualization of Results:
-	Adding more graphs and visual examples of parameter recovery would make the results more illustrative.
6.	Revise the article in terms of increasing the volume to the required one.
Conclusion:
The article makes a significant contribution to the fields of acoustic tomography and deep learning. The proposed method demonstrates high potential for solving complex inverse problems. However, to increase the scientific value and practical applicability of the work, it is recommended to refine certain aspects, such as validation on real-world data, improving the handling of non-convex shapes, and adding comparisons with other methods. Revise the article in terms of increasing the volume to the required one. After these improvements, the article could be recommended for publication in a high-ranking scientific journal.

---

### Official Review · Reviewer_xfr8 · 2025-02-25
**The article presents a significant contribution to the field of acoustic tomography with the introduction of a two-stage data processing method for recovering the absorption parameter. The combination of a physically accurate mathematical model with a deep learning-based approach is innovative and well-executed. The mathematical derivations are rigorous, and the use of synthetic data for training and validation is appropriate. However, the article could benefit from real-world validation, a more detailed analysis of computational complexity, and improvements in handling non-convex shapes. Overall, the proposed method is a promising advancement in acoustic tomography, and the article provides a solid foundation for future research in this area.**

**Rating:** 8
**Confidence:** 2

**Review:**

The article presents a novel approach to recovering the acoustic absorption parameter in acoustic tomography using a two-stage data processing method. The authors propose a deep learning-based framework that combines a physically accurate mathematical model with a neural network to reconstruct the absorption distribution from acoustic pressure data. The method is designed to overcome the challenges associated with traditional model-driven approaches, particularly the difficulty in recovering absorption due to its weak influence on the data. The article is well-structured, neural network architecture is described, but the abstract is not detailed enough.

Strengths of the Article:

1. Approach:
   The article introduces a two-stage data processing method that first computes the gradient of the cost functional using a physically accurate mathematical model and then uses a neural network to recover the absorption distribution. This approach leverages the strengths of both model-driven and data-driven methods, making it a significant contribution to the field of acoustic tomography. The use of a hyperbolic first-order system for wave propagation is particularly noteworthy, as it provides a more realistic physical model compared to traditional wave equation-based methods.

2. Mathematical Rigor:
   The article is mathematically rigorous, with detailed derivations of the hyperbolic system of partial differential equations (PDEs) used to model the acoustic wave propagation. The authors also provide a thorough explanation of the adjoint problem and the gradient computation, which are essential for solving the inverse problem. The mathematical foundation of the method is solid, and the derivations are well-presented.

3. Synthetic Data Generation:
   The authors generate a synthetic dataset to train and validate the model, which is a crucial step in ensuring the robustness of the proposed method. The dataset includes a variety of inclusion shapes and absorption values, allowing for a comprehensive evaluation of the model's performance. The use of synthetic data also enables the authors to control the data generation process, ensuring that the training examples are diverse and representative of real-world scenarios.

4. Neural Network Architecture:
   The article employs a UNet convolutional neural network architecture, which is well-suited for image-related tasks such as segmentation and reconstruction. The detailed description of the encoder, bottleneck, and decoder components of the UNet architecture is clear and informative. The use of techniques such as batch normalization, dropout, and transposed convolution demonstrates the authors' understanding of modern deep learning practices.

5. Computational Results:
   The computational results presented in the article are promising, showing that the model can successfully reconstruct the absorption distribution from the gradients of the cost functional. The visualization of the reconstructed images demonstrates a high degree of overlap with the true absorption distribution, particularly for convex shapes. The article also provides a detailed analysis of the model's performance on different types of inclusions, including non-convex shapes, which adds depth to the evaluation.

Weaknesses and Areas for Improvement:

1. Limited Real-World Validation:
   While the synthetic dataset is well-constructed, the article lacks validation on real-world data. The performance of the proposed method on experimental or clinical data would provide stronger evidence of its practical utility. Real-world data often contain noise, artifacts, and other complexities that are not fully captured by synthetic data, and testing the method on such data would be a valuable addition.

2. Computational Complexity:
   The article does not provide a detailed analysis of the computational complexity of the proposed method. Given that the method involves solving both direct and adjoint problems, as well as training a deep neural network, it is important to evaluate the computational cost and scalability of the approach, especially for large-scale or real-time applications.

3. Performance on Non-Convex Shapes:
   The article acknowledges that the model performs poorly on non-convex shapes, which is a significant limitation. The authors suggest that this may be due to the complexity of recognizing non-convex patterns, but they do not provide a detailed analysis or potential solutions. Further investigation into improving the model's performance on non-convex shapes would enhance its applicability.

4. Comparison with State-of-the-Art Methods:
   The article does not include a comprehensive comparison with other state-of-the-art methods for recovering absorption in acoustic tomography. While the authors mention related works in the introduction, a direct comparison with existing methods would provide a clearer understanding of the proposed method's advantages and limitations.

5. Generalization to 3D:
   The article focuses on 2D acoustic tomography, but many real-world applications require 3D reconstruction. Extending the proposed method to 3D would be a valuable contribution, as it would allow for more accurate and comprehensive imaging of complex structures.

Detailed Comments:

1. Mathematical Model:
   The mathematical model based on the first-order hyperbolic system is well-formulated and appropriate for describing acoustic wave propagation. However, the article could benefit from a more detailed discussion of the assumptions and limitations of the model, particularly in the context of real-world applications where the medium properties may be more complex.

2. Data Generation:
   The synthetic data generation process is well-described, but the article could provide more details on the specific parameters used for generating the dataset, such as the range of absorption values, the size and shape of the inclusions, and the spatial resolution of the images. This would help readers better understand the scope of the dataset and the challenges faced during training.

3. Neural Network Training:
   The training process is described in detail, including the use of the Adam optimizer and the MSE loss function. However, the article could provide more insights into the choice of hyperparameters, such as the learning rate, batch size, and the number of epochs. Additionally, the authors could discuss any challenges encountered during training, such as overfitting or convergence issues, and how they were addressed.

4. Computational Results:
   The computational results are presented clearly, with visualizations of the reconstructed images. However, the article could benefit from a more quantitative analysis of the results, such as the use of metrics like peak signal-to-noise ratio (PSNR) or structural similarity index (SSIM) to evaluate the quality of the reconstructions. This would provide a more objective assessment of the model's performance.

5. Future Work:
   The article outlines several promising directions for future research, including the extension of the method to 3D and the improvement of performance on non-convex shapes. These are valuable suggestions, and pursuing these directions could further enhance the applicability and robustness of the proposed method.

Conclusion:

The article presents a significant contribution to the field of acoustic tomography with the introduction of a two-stage data processing method for recovering the absorption parameter. The combination of a physically accurate mathematical model with a deep learning-based approach is innovative and well-executed. The mathematical derivations are rigorous, and the use of synthetic data for training and validation is appropriate. However, the article could benefit from real-world validation, a more detailed analysis of computational complexity, and improvements in handling non-convex shapes. Overall, the proposed method is a promising advancement in acoustic tomography, and the article provides a solid foundation for future research in this area.

Recommendation:

I recommend the article for publication, provided that the authors address the aforementioned weaknesses and areas for improvement. Specifically, the authors should consider validating the method on real-world data, providing a more detailed analysis of computational complexity, and improving the model's performance on non-convex shapes. Additionally, a comprehensive comparison with state-of-the-art methods would strengthen the article's contribution to the field. With these improvements, the article will be a valuable addition to the literature on acoustic tomography.

---

### Official Review · Reviewer_BYJ9 · 2025-02-26
**This paper is dedicated to the reconstruction of the acoustic parameters of the medium the acoustic absorption by using the pressure. The inverse problem has been solved. CNN was developed and tested on synthetic data.**

**Rating:** 7
**Confidence:** 4

**Review:**

The mathematical model for solving the motion of an acoustic wave in different media by the following first order system of hyperbolic equations is described. A synthetic dataset was created to train and validate the model. The architecture of the UNet convolutional neural network is used to solve the problem of restoring the density of the medium by spatial gradients. After training the model on a synthetic dataset, testing was performed to assess the quality of the recovering of acoustic absorption. The model successfully reconstructed the density of the medium from the gradients of space. The metrics RMSE, MAE have been used for testing.

Strong
- Paper describes the approach and provide experimental results obtained on synthetic data from mathematical model
- The authors developed their own program code

Weak
- Paper does not explain practical applications of the used method
- Paper does not compare results with alternative approaches

Comments should be addressed to improve the content of the article.
1.	In Fig.1 it is not explained what arrows 6, 7 are used for.
2.	In equations 5-8 and further it is not explained what the auxiliary functions Ψ are.
3.	It is not given how the δ-function was approximated in the numerical solution of the problem.
4.	The data in Table 1 are not clear. Are these values good? To which problems are they relevant?
5.	The section “ACKNOWLEDGMENTS” should be changed.

---

### Decision · Program_Chairs · 2025-03-08

**Decision:**

Accept (Oral)

**Comment:**

Your article has been accepted and you can give a talk on the article. All articles will be sorted by rating and within the available conference places one author from each article will be invited. If there are not enough places, then you will either have the opportunity to speak remotely or come at your own expense!